# Implementation of a Provincial Long COVID Care Pathway in Alberta, Canada: Provider Perceptions

**DOI:** 10.3390/healthcare12070730

**Published:** 2024-03-27

**Authors:** Kiran Pohar Manhas, Sidney Horlick, Jacqueline Krysa, Katharina Kovacs Burns, Katelyn Brehon, Celia Laur, Elizabeth Papathanassoglou, Chester Ho

**Affiliations:** 1Neurosciences, Rehabilitation and Vision, Strategic Clinical Network, Alberta Health Services, 10301 Southport Lane SW, Calgary, AB T2W 1S7, Canada; jkrysa@ualberta.ca (J.K.); papathan@ualberta.ca (E.P.); chester.ho@ahs.ca (C.H.); 2Community Health Sciences, Cumming School of Medicine, University of Calgary, Calgary, AB T2N 1N4, Canada; 3Faculty of Nursing, University of Alberta, Edmonton, AB T6G 1C9, Canada; 4School of Public Health, University of Alberta, Edmonton, AB T6G 1C9, Canada; horlick@ualberta.ca (S.H.); katharina.kovacsburns@albertahealthservices.ca (K.K.B.); 5Faculty of Rehabilitation Medicine, University of Alberta, Edmonton, AB T6G 2E1, Canada; brehon@ualberta.ca; 6Department of Clinical Quality Metrics, Alberta Health Services, Edmonton, AB T5J 3E4, Canada; 7Women’s College Hospital Institute for Health System Solutions and Virtual Care, Toronto, ON M5S 1B2, Canada; celia.laur@wchospital.ca; 8Institute of Health Policy, Management and Evaluation, University of Toronto, Health Sciences Building, Toronto, ON M5T 3M6, Canada; 9Division of Physical Medicine and Rehabilitation, Faculty of Medicine & Dentistry, University of Alberta, Edmonton, AB T6G 1C9, Canada

**Keywords:** long COVID, screening, service navigation, self-management, implementation, qualitative

## Abstract

A novel, complex chronic condition emerged from the COVID-19 pandemic: long COVID. The persistent long COVID symptoms can be multisystem and varied. Effective long COVID management requires multidisciplinary, collaborative models of care, which continue to be developed and refined. Alberta’s provincial health system developed a novel long COVID pathway. We aimed to clarify the perspectives of multidisciplinary healthcare providers on the early implementation of the provincial long COVID pathway, particularly pathway acceptability, adoption, feasibility, and fidelity using Sandelowki’s qualitative description. Provider participants were recruited from eight early-user sites from across the care continuum. Sites represented primary care (n = 4), outpatient rehabilitation (n = 3), and COVID-19 specialty clinics (n = 2). Participants participated in structured or semi-structured virtual interviews (both group and 1:1 were available). Structured interviews sought to clarify context, processes, and pathway use; semi-structured interviews targeted provider perceptions of pathway implementation, including barriers and facilitators. Analysis was guided by Hsieh and Shannon as well as Sandelowski. Across the eight sites that participated, five structured interviews (n = 13 participants) and seven semi-structured interviews (n = 15 participants) were completed. Sites represented primary care (n = 4), outpatient rehabilitation (n = 3), and COVID-19 specialty clinics (n = 2). Qualitative content analysis was used on transcripts and field notes. Provider perceptions of the early implementation outcomes of the provincial long COVID pathway revealed three key themes: process perceptions; awareness of patient educational resources; and challenges of evolving knowledge.

## 1. Introduction

A novel, complex chronic condition emerged from the COVID-19 pandemic: long COVID [1,2,3,4,5]. While there are challenges with inconsistent definitions of long COVID [5], the World Health Organization characterizes long COVID as the “continuation or development of new symptoms 3 months after the initial SARS-CoV-2 infection, with these symptoms lasting for at least 2 months with no other explanation” [6]. A 2022 systematic review (n = 194 articles) and meta-analysis (n = 735,006 participants) found that the pooled prevalence of COVID-19 survivors experiencing ≥1 unresolved symptom, regardless of hospitalization status, was 45%, while that for hospitalized and non-hospitalized participants was 52.6% and 34.5% (where the mean follow-up period was 126 days) [7]. In Canada, the prevalence of longer-term symptoms is 17.2% (95% CI: 15.8%, 18.7%) based on a national 2022 survey [8].

Symptoms can be multisystem and variable, with nearly 200 reported by patients [9,10], affecting the lungs, kidneys, digestive tract, brain, heart, and other organs [5]. Fatigue, pain/discomfort, impaired sleep, breathlessness, and impaired usual activity are the most prevalent [7]. Long COVID may present episodically, with symptoms disappearing or remitting for a period before relapse occurs [9,11]. The experience of long COVID consistently includes significant functional impairment as well as impacts on quality of life and return-to-work [12,13,14]. The latter are exaggerated for persons who were hospitalized during acute infection versus those who were never hospitalized [12]. 

Effective long COVID management requires multidisciplinary, collaborative models of care, which continue to be developed and refined [15]. Evidence demonstrates that rehabilitation may improve symptoms [16,17,18,19]; however, evidence-based rehabilitation models for long COVID are currently lacking [15]. A review of pan-Canadian models of care for long COVID noted that routine dismissals from providers contributed to frequent misdiagnoses; a national roundtable called for better diagnostic processes to reduce misdiagnosis and increase access to care [20]. Healthcare providers and patients with long COVID (n = 43) in the UK recommended that long COVID service models should be integrated, multi-system-focused, armed with multiple specialties, and geared towards the uncertainty inherent in long COVID [21]. Examples of the multidisciplinary rehabilitation management of long COVID include tiered service models that involve primary care, rehabilitation providers, specialist care, and/or self-management resources. Developing effective models of care in complex, adaptive health systems can be challenging and time-consuming. Early insights can be garnered from understanding the acceptability and perspectives of providers who were early users of these long COVID models of care. Understanding providers’ perspectives and challenges can inform use in more locations and ensure that early challenges are not replicated by subsequent users. Only a few studies have clarified provider perspectives on delivering long COVID rehabilitation [22]. For example, a study primarily composed of providers who have long COVID spoke as much to patient experience as provider experience, and in the context of few formal models of care [21]. 

### Context

Alberta is one Canada’s 10 provinces; it has the largest provincial, fully integrated, single-payer healthcare system in Canada, which delivers healthcare to 4.4+ million people [23]. Between 2020 and 2022, a collaborative, provincial, consensus-based approach involving 129 multidisciplinary stakeholders (in a long COVID taskforce) was used to develop a novel care pathway for the screening, triage, and management of long COVID (the Post-COVID Rehabilitation Framework) [24]. The pathway performs the following:Prescribes use of specific symptom screening (adapted from the COVID-19 Yorkshire Rehab Screen Tool) [25] and assessment tools (Post-COVID-19 Functional Status Scale (PCFS)) [26];Outlines care pathways based on the PCFS score to align with the intensity of rehabilitation needs;Provides self-management and educational resources for patients and providers [24].

The available rehabilitation in Alberta, in order of worsening functional impairment, included universal self-management resources, group programs, and 1:1 rehabilitation [24]. 

Discrete implementation strategies were used by the provincial long COVID taskforce to support the co-design and early dissemination of the rehabilitation pathway between 2021 and 2022, which included (a) building a coalition of stakeholders to advocate for implementation; (b) identifying and training (local and system) champions; (c) developing and distributing educational materials; (d) translating patient resources into several languages (i.e., Arabic, Cantonese, Cree, French, Punjabi); (e) promoting network weaving to incorporate the strategy into existing information-sharing relationships; (f) promoting adaptability (e.g., modifying provider material to reflect local realities); and (g) involving patients and families in pathway development and realization. Pathway uptake was not mandatory but highly recommended. While the uptake of the pathway has been variable, the receipt, perception, and impact of these implementation strategies amidst the dynamic post-COVID-19 pandemic landscape across Alberta remain unclear. 

We sought to clarify the perspectives of multidisciplinary healthcare providers working at early-user sites of Alberta’s long COVID rehabilitation pathway on (a) the acceptability (i.e., perceptions of the pathway); (b) adoption (i.e., intention to use the pathway); (c) feasibility (i.e., barriers and facilitators to pathway implementation); and (d) fidelity (i.e., level of adaptation of the pathway upon implementation) of pathway implementation and implementation strategies [27].

## 2. Materials and Methods

This study received ethics approval from the Health Research Ethics Board of the University of Alberta (Pro00113182). The study methodology involved qualitative description, given its ability to completely describe and explain a phenomenon of interest within healthcare settings using a variety of methods [28,29,30]. 

### 2.1. Settings, Participants, and Recruitment

Recruitment of sites and participants occurred via purposeful sampling. The research team recruited early-user sites through presentations and engagement with the provincial long COVID taskforce and working groups, which included healthcare operational leaders involved in long COVID care provision. Known intermediaries provided study overviews, and interested leaders contacted the research team to begin discussions on participation.

Study sites were healthcare clinics from across Alberta, providing, or planning to provide, care for long COVID patients. Sites self-identified as early users currently implementing, or intending to implement, some or all of the provincial long COVID rehabilitation pathway in their care processes. We sought maximum variation in site recruitment across the care continuum from primary care, medical specialties, and outpatient rehabilitation (including outpatient clinics and telehealth delivery), as well as across geographically diverse settings (metropolitan urban, regional urban, and rural areas) and different care delivery methods (in-person vs. telehealth). 

The study participants included multidisciplinary healthcare providers working at early-user study sites. Role-wise, we sought maximum variation across front-line clinical staff, clinical team leads, site managers, and leadership. Discipline-wise, we sought maximum variation across medicine, nursing, occupational therapy, physiotherapy, and other allied health disciplines. There were no exclusion criteria for providers. Data collection was planned until data saturation, or when three repeated attempts at recruitment at the same site yielded no further participants.

Once site leaders were onboarded, managers shared study overviews and researcher emails with their clinical staff. Interested individuals reached out to the researchers directly, whereafter informed consent was discussed and study participation was organized.

### 2.2. Data Collection

Data collection was two-fold to achieve a comprehensive understanding of the long COVID rehabilitation pathway acceptability (i.e., perceptions of the pathway), adoption (i.e., intention to use the pathway), feasibility (i.e., barriers and facilitators to pathway implementation), and fidelity (i.e., level of adaptation of the pathway upon implementation) in diverse Alberta early-user sites. First, structured interviews (by researchers SH or JK) with site leads set out the process and implementation characteristics of the site context. Second, semi-structured interviews (by researchers SH or JK) with front-line providers (1:1 and group) clarified the provider perspectives on the interim implementation outcomes of the long COVID rehabilitation pathway. Both researchers SH and JK are health-research trainees, with previous education and experience in trained qualitative research data collection and analysis.

#### 2.2.1. Structured Interviews 

One-on-one, virtual, structured interviews were completed with consenting early-user site leads, including managers. Closed-ended questions guided the 15–30 min interviews (Appendix A) [31,32]. Questions sought clarity on site human resources, patient population, referral processes, assessment and diagnosis procedures, as well as perceived pathway acceptability, adoption, feasibility, and fidelity. While one researcher conducted the interview, another took detailed field notes of the responses during each structured interview. All interviews were audio-recorded to verify and supplement the field notes. Structured interview responses were directly recorded verbatim without using transcripts. Due to limited time with the clinician participants, demographic questions had to be removed from the structured and semi-structured interview guides.

#### 2.2.2. Semi-Structured Interviews

The data retrieved from closed-ended questions complemented, and provided context for, the open-ended question responses. Informed by the structured interviews, virtual individual or group semi-structured interviews were completed with consenting front-line clinicians (Appendix A). Open-ended questions queried current roles and responsibilities; current approaches and challenges to screening and assessing for long COVID symptoms; training and perceptions of the PCFS (as a proxy of framework use); familiarity and understanding of the provincial long COVID rehabilitation pathway; familiarity with and understanding of the available educational resources for patients or providers; and perceived acceptability, adoption, feasibility, and fidelity of the pathway. All semi-structured interviews were audio-recorded and confidentially transcribed verbatim. Written field notes were recorded during the interview. Due to limited time with the clinician participants, demographic questions had to be removed from the structured and semi-structured interview guides.

### 2.3. Data Analysis 

Structured responses, transcripts, and field notes for both the structured and semi-structured interviews were collected and analyzed concurrently using Microsoft Excel 365 and NVivo 12, respectively. The different data sources were deemed complementary, offering contextual insights or content details, respectively. An audit trail was kept of decision-making throughout the interview and analysis process. The structured responses were descriptively analyzed with a focus on frequencies, distributions, and key contextual practices. Meanwhile, qualitative content analysis was employed to identify commonalities and differences in the semi-structured transcripts and field notes [28,33,34]. Field notes were inductively coded to identify sub-categories within these broad categories. Semi-structured interview transcripts were analyzed thematically. One interviewer (SH) and one independent member of the research team (KB) independently coded each interview. A third researcher (JAK) independently assessed the coding to resolve discrepancies and ensure alignment between the codes and transcripts. The researchers did not have previous connections with the participants. Team members met regularly to develop and review the coding framework, which sought to expose relationships between themes and to produce a holistic understanding of the participants’ perspectives. Thick description with detailed participant quotes substantiated the coding framework.

## 3. Results

### 3.1. Context: The Early-User Sites

Contextually, these interviews provided insights into long COVID care processes and management across these early-user sites. All sites were specific to adult patients who faced a wide range of functional impairments; the outpatient rehabilitation sites limited their patient population to those with active rehabilitation needs. Of the primary care sites, four were situated in primarily rural settings, and one serviced a major metropolitan area. Of the outpatient rehabilitation sites, one was a provincial telerehabilitation service, while two serviced different major metropolitan areas. Of the medical specialty clinics which focused on long COVID, both serviced the same major metropolitan area.

All sites were multidisciplinary but varied in their workforce. The professionals available across primary care (not just those who participated in the interviews), outpatient rehabilitation and medical specialty clinics included physiotherapists (n = 4), occupational therapists (n = 4), physicians (n = 3), a dietitian (n = 1), licensed practical nurses (n = 2), nurse practitioners (n = 2), psychologists (n = 2), recreation therapists (n = 2), registered nurses (n = 2), social workers (n = 2), speech language pathologists (n = 2), therapy assistants (n = 2), and a pharmacist (n = 1). 

### 3.2. Participants

Between January and August 2022, eight early-user sites supported study recruitment from their teams, leading to five structured, team-lead interviews (n = 13 participants; four group and one 1:1 interviews) and seven semi-structured, multidisciplinary, front-line clinician interviews (n = 15 participants; five group and two 1:1 interviews). These sites represented primary care (n = 4), outpatient rehabilitation (n = 3), and COVID-19 specialty clinics (n = 2) (Table 1). The structured and semi-structured interviews ranged from 30 to 60 and 20 to 60 min, respectively. Not all early-user sites participated in both the structured and unstructured interviews.

In the structured interviews, the role participants primarily identified with included team leads (n = 6), senior consultants/advisors (n = 2), physiotherapists (n = 2), a director (n = 1), a physician (n = 1), and an occupational therapist (n = 1). The professions represented in the semi-structured interviews included nursing, pharmacy, physiotherapy, occupational therapy, and recreation therapy. Because recruitment was facilitated through intermediaries (team leads) to protect clinician privacy, we are unable to determine how many people declined study participation and their reasons therein.

#### Referral Pathways and Screening Tools

Referral processes, and thus access, varied across the sites. At the four primary care and single provincial outpatient telerehabilitation sites, no referral was required to access services. At the two geographically specific outpatient rehabilitation sites, referral processes varied. One outpatient rehabilitation site required two criteria for referral: (i) >12 weeks post-acute infection and (ii) a PCFS score of 3 or 4 (signaling moderate to severe functional impairment). The other outpatient rehabilitation site required referrals from family physicians or other healthcare providers. At the two medical specialty sites, referrals were required from family physicians, emergency departments, or other healthcare providers. 

The screening tools used for suspected long COVID varied across all early-user sites, despite the proposed standard long COVID provincial rehabilitation pathway. The primary care clinics had no long-COVID-specific pathway or plans at the time of data collection; long COVID patients were treated similarly to all others. At the provincial outpatient telerehabilitation early-user site, three tools were consistently used: PCFS, EuroQol-5D-5L (quality of life), and the Post-Viral Fatigue Screening tool. To screen rehabilitation needs in persons with long COVID at the specialty medical clinic focused on long COVID, the PCFS was primarily used, alongside the identification of the top three most impactful symptoms. The two geographically specific outpatient rehabilitation early-user sites took different approaches to long COVID screening. One site focused exclusively on the Canadian Occupational Performance Measure (COPM), as used for all patients visiting that site. The other site completed a comprehensive long-COVID-specific screening of physical, cognitive, communication, psychosocial, general function, activities of daily living, and life roles.

### 3.3. Provider Perceptions of the Pathway 

Provider perceptions of the early implementation outcomes of the provincial long COVID rehabilitation pathway revealed the following key themes: process perceptions; awareness of patient educational resources; and challenges of evolving knowledge. Each theme had three sub-themes, elaborated on below. The sub-themes were often interconnected.

#### 3.3.1. Process Perceptions

The sub-themes of process perceptions included perceived utility, inconsistent utilization, as well as relative advantages and disadvantages.

#### 3.3.2. Perceived Utility

Providers described several benefits of using a rehabilitation pathway for the care of long COVID patients. Providers felt that the pathway was beneficial for the identification and streaming of long COVID patients, for triage and resource management, and for giving patients a better chance to reach the appropriate services and to feel supported. Pathway use benefited providers by facilitating the provision of equitable care by providing some uniformity of how care is delivered and keeping resources at the forefront to help providers find them easily. Some providers perceived the following disadvantages: pathways were felt to take significant time to update and were not always up-to-date with practice; provider unfamiliarity with the screening tool may have excluded some patients from receiving appropriate care; and the current pathway was not visually intuitive and required time to digest for some providers. 

“*I do feel like you do get a pretty accurate … global [score]. Kind of idea at least of where the person is at.*”(Interview 1).

“*I think it gives you a global score, which I’m sure is why it was … set out [sic] across the province useful one.*”(Interview 2).

“*I find it really helpful. … I say kind of categorize those patients and, and direct them to the right resources; but also, [sic] I use it as a discussion tool with a patient.*”(Interview 7).

“*[A disadvantage) is that if a provider isn’t familiar with the tool and isn’t using the tool, then those patients are maybe not able to access the resources, are—are not aware that the resources exist. And so consequently are not being connected with those resources if they are experiencing more severe symptoms, especially those intensive resources.*”(Interview 7).

“*I mean mindboggling when you try to read it, but you know maybe a good set of glasses, cause it’s pretty tiny.*”(Interview 9).

#### 3.3.3. Inconsistent Utilization

The structured interviews revealed that the provincial pathway (and its inherent resources and tools) was inconsistently applied across early-user sites. Each site had varying methods of identifying post-COVID patients and determining appropriate resources and/or referrals for each patient. Sites that fully implemented the pathway generally used the PCFS in day-to-day practice. Those that partially implemented the pathway used the PCFS solely as part of their criteria for accepting a referral or complemented the PCFS with additional tools. For the latter sites, the PCFS was used to determine patient trajectory along the pathway; others used additional or alternate screening tools to determine functional impairment and the services required for that patient. It was noted that greater hesitation to adopt the PCFS was described by front-line clinicians versus leadership.

#### 3.3.4. Relative Advantages and Disadvantages

Noted advantages of the PCFS included the introduction of a standard benchmark to trigger further discussions; a quantitative measure to hold onto; a discussion tool with patients; and a measure of insight into the patients’ conditions beyond long COVID.

“*It gives a benchmark. It gives us like that point of okay here’s where we’re going to start. I would be interested to know like [what] [sic] clinically significant difference. … When you’re doing more callbacks, we’ll actually readminister it.*”(Interview 2).

“*Which is kind of cool cause then you can be like “oh you went from a three to a two, like great”. It’s a nice kind of quantitative measure.*”(Interview 2).

“*I use it as a discussion tool with a patient. So when we’re talking about their symptoms and what the impact of their symptoms are on their daily lives, that helps me to sort of determine how impacted they are, which also impacts the conditions we’re treating them for separate from COVID.*”(Interview 7).

Some providers perceived the PCFS as being insufficiently specific and queried its reliability and utility. Some early-user site providers’ perceptions on the utilization (structured interviews) are described in Table 1. 

Some participants felt that the PCFS did not provide enough information to inform clinical decision-making and care planning. The global, discrete grading of the PCFS led to perceived challenges with providing an accurate picture of functional impairment, especially when self-scored. Some providers described disparities between the patient’s PCFS recorded at referral versus the patient’s functional impairment at the first appointment. Some providers described how, at times, patients were provided inappropriate referrals for their level of functional impairment due to the inaccuracy of their PCFS score. Conversely, some clinicians were concerned that the PCFS score requirement for referral to rehabilitation could lead to patients falling through the cracks. Some clinicians were concerned by a perceived “medical model” orientation of the PCFS related to its symptom checklist, which deterred some allied health professionals from using it. The value of standardized screening and using a consistent tool was recognized by providers, but the noted concerns outweighed the perceived value.

Of note, generally, providers from primary care sites found greater utility of the PCFS to assist with patient referrals to specialty clinics for more in-depth assessment and treatment. Conversely, rehabilitation care providers described concerns regarding the use of the PCFS as the sole assessment tool for patients experiencing long COVID symptoms. These concerns included the perceived difficulty for others and for themselves to complete the PCFS; uncertainty around how and when to administer and re-administer PCFS; perceived unreliability of PCFS scores from referrals sources; and risk of patients not receiving needed care due to inaccurate scores. These concerns resulted in many rehabilitation sites implementing the use of additional, or alternate, long COVID tools. This use of other tools countered the rehabilitation pathway’s intention for the PCFS to be a screening, not an assessment, tool.

As highlighted in Table 1, data collection and storage varied for the PCFS tool scores. This was complicated by a phased rollout of a provincial electronic medical record (i.e., phased into some but not all early-user sites). Some sites were still using paper charts. This variation impacted communication between referral sites, and the standardization of data collection and storage.

“*My only unfortunate thought when it comes to pathways, is that they take sometimes too long to actually go with…the changes that are occurring. So if we’re going to have those pathways we need them to be authored by people who are in the know, are on the frontlines and see the changes as they’re coming and can make those changes to adjust as we’re going.*”(Interview 6).

“*The reason we use the Yorkshire—it’s very similar to the PCFS—except that I don’t find that you can gather enough information from just using the PCFS. And so this tool incorporates the PCFS, but it also has additional questions or areas to explore that would probably identify more issues with the patient.*”(Interview 6).

“*[The pathway and the PCFS] kind of groups [patients] into … one category, it doesn’t really take them as a whole. So you can kind of—you can ask the questions about cognition and you can ask the questions, let’s say … the PESE scale, but it doesn’t take into consideration of them as a whole. … I just think it groups them into one and then you’re with a scale, [so] maybe they don’t get the right … care that they need.*”(Interview 10).

“*I don’t know that I ever really feel like [the PCFS] captures what is really happening with the person. So there [is] three or four, [but] there’s a lot of gradient between three and four … [A]nd to refer them on most places require … PCFS of three and … so if they’re a two, then right away we’re thinking … so now what? … [A]nd it doesn’t mean that they’re back to all their previous functional levels, right? The screening and the assessment I find is just maybe too generic for … my personal needs as a therapist.*”(Interview 1).

#### 3.3.5. Awareness of Patient Educational Resources 

The sub-themes of awareness of patient educational resources related to patient needs, content areas to cover, and the necessity for adaptation for individual patients.

Awareness of the available provincial long COVID resources was central to their access and use. Providers reported that many patients did not know about the available resources and support for long COVID recovery in Alberta. Providers felt there was a need to raise awareness among patients about these resources and noted that, in many cases, patients were not receiving this education from their primary care provider. Providers spoke of how a significant proportion of their care for long COVID patients involved the provision of education and resources to help patients recognize and manage symptoms. The most-used resources were information on pacing (energy management), breathing exercises, and coping with dizziness.

“*I would say most patients are aware of long COVID, I think it’s been … in the media [enough] that patients are aware this is a thing; but they don’t seem to recognize that there are resources and places that they can go for support, other than accessing them through their primary care provider which is certainly the right thing to do … but it seems to stop there.*”(Interview 7).

“*The main way providers support patients is through education. With symptoms so varied, teaching patients about long COVID, and how to manage symptoms, is highly important to recovery. It’s really just trying to get them to recognize the symptoms of their body and to respect that okay, my body is telling me no.*”(Interview 1).

Providers generally found the available resources useful for their patients, especially those who did not require one-on-one, specialized care. Providers appreciated the centralization of resources online as it facilitated ease of navigation to the appropriate resources. With energy management being a significant challenge for many long COVID patients, many standardized resources did not initially meet patients’ needs; they were too lengthy for patients with brain fog and fatigue to navigate. Providers described paring down the recommended resources or directing patients to look at only specific sections of documents, as the original form of these resources was overwhelming for their patients. 

“*I think that, having these resources, having a place where they can be found, having resources that are accessible to patients without having to access specialized care [has been done well]. Like some of the videos the website and places where they can go in and look and get information, I think has been done really well.*”(Interview 7).

“*We send a self-management document…it’s overwhelming cause it is 60 pages. So we’ve learned to say, you know just look at one symptom, look at a symptom that you know you have. You have fatigue, look at that area, just click on that one. And—and so like don’t try and read the whole thing, cause I’ve tried to read the whole thing and it’s mindboggling, so if your brain’s not working right it’s going to be even worse.*”(Interview 9).

#### 3.3.6. Challenges of Evolving Knowledge

The sub-themes of challenges of evolving knowledge related to time, ever-changing information, and lack of awareness.

Some providers found it difficult to keep up with the continuous influx of information on long COVID and simultaneously manage high patient volumes. Manageable caseloads were associated with greater confidence with, and use of, PCRF tools and resources. In those cases, providers described having more time to learn and understand. Team capacity may act as a mitigating factor for providers with limited time, as providers described how team members are able to support each other to filter through information and provide the best available resources to patients.

“*I sort of came into this position … and literally my feet hit the ground running, and so I have only had an opportunity to keep up with what I can do for my patients, how I can help my patients; what I can do to provide for my patients. I haven’t had an opportunity to look at what education is provided to physicians.*”(Interview 6).

“*We had a lot of information coming at us, but I think it was dealt with well, that we were given the time as clinicians to maybe go to—go to these learnings opportunities and talk to other clinicians.*”(Interview 10).

Many providers were unaware of the long COVID rehabilitation pathway, though these providers were using other types of processes for screening and directing patients to services and resources. Many providers expressed less awareness of other aspects of the pathway, such as educational resources. In other cases, providers were aware of the pathway, but frequent updates made it difficult to keep it up-to-date. This contributed to reduced confidence in providers regarding its use. Providers’ capacity to handle evolving and emerging information seemed critical to the feasibility and adoption of the rehabilitation pathway and the experience of supporting persons with long COVID.

Some providers described their difficulty with navigating the health system website housing the COVID-19 resources. This, in conjunction with the above-noted lack of time to become familiar with new or updated resources, made it more difficult for providers to stay up-to-date with how the AHS sought to treat and manage long COVID. Some providers defaulted to directing patients to similar, familiar resources even as information evolved. This also links to provider education on the pathway: poor ease of access was a contributor to the observed lack of provider knowledge on the pathway and the available resources. 

“*I think [I’m] moderately familiar [with the pathway]. I think it’s … always changing, so I would say at the moment like not as confident as I was, because I haven’t used it for a bit. … So like at the moment not very confident cause I just learned about [the pathway] … like ten minutes before talking to you*”(Interview 2).

“*When I click on our page here and this page has evolved so much… And it can be a bit overwhelming to navigate as a healthcare provider…. [I feel like] wow, a lot has changed, this is a bit overwhelming, where do I start?*”(Interview 2).

“*[What prevents use of resources is] knowing where they are and what’s available, like again just like the, the vast quantity. I think there’s a ton out there, but just I think for me, like kind of re-immersing myself in that refamiliarizing myself, and being able to—to know what’s out there. I think that’s a barrier, sometimes knowing what’s out there, because you kind of go to your default—to the ones that you’ve used.*”(Interview 2).

## 4. Discussion

The implementation of a novel provincial pathway for long COVID rehabilitation faced three categories of barriers and facilitators: process perceptions; awareness of patient educational resources; and challenges of evolving knowledge. While there was agreement on the need for standardized resources for patients and providers, as well as screening tools to recognize and appropriately manage long COVID, multidisciplinary providers were challenged in terms of fully adopting the pathway. Pathway adoption during the study period was highly variable. There seemed to be confusion in the aim of using the PCFS as a standardized tool in the pathway; while developed as a screening tool, it was understood by some as an assessment tool. Those with the latter understanding then became resistant to its utilization and found alternatives. Resources had to be modified to improve acceptability, which reduced fidelity. Shifting characteristics and dynamic forces challenged clinicians’ perceptions of the pathway’s feasibility, most importantly the consistently evolving understanding on this novel condition and lack of time to keep up-to-date. Many negative perceptions on the process pieces and tools of the pathway impeded the PCFS’s acceptability and fidelity particularly. This had equity implications as a common rehabilitation pathway was developed for provincial adoption, but due to implementation challenges, unstandardized, variable practices emerged. Persons with long COVID across Alberta faced divergent experiences based on the providers they visited. A review of the pathway development process [24] reveals that few front-line clinicians were involved in that process; this gap may have contributed to the perceived barriers recognized in this study and represent an area for improvement in future pathway refinement or development.

Recommendations emerge for this context and similar contexts seeking to implement long COVID rehabilitation care pathways involving multidisciplinary teams; sites across the care continuum; standardized tools for long COVID screening and assessment; and diverse educational resources for patients and providers. First, a dedicated communication strategy for provincial dissemination of the self-management resources, especially to primary care, could promote the adoption and feasibility of the pathway. These may be complemented by educational webinars hosted by professional and organizational communities (e.g., medical association, primary care networks). Second, the creation of a communication pathway or community of practice between geographically disparate long COVID teams would facilitate information sharing and interprofessional collaboration. Third, the long COVID pathways may need to be clarified and updated to confirm the required standardized processes versus optional complementary functional screening and assessment tools. The required pathway components could also be highlighted and embedded in the organization-wide electronic medical record for the consistency of notification and access. This could advance the fidelity and feasibility of the required components of the pathway. 

The pathway developed likely requires greater emphasis and strategies on interprofessional communication and collaboration to empower teams to share information and to trust that shared information. This could reduce the duplication of screening conducted on patients with significant fatigue issues; allay concerns about patients falling through the cracks; and foster the standardization of care and access across diverse sites. A consistent health system challenge is the quality of transitions across the care continuum. This is reflected in organizational health system priorities, such as in Alberta, where these are patient flow [23] and transitions between home and hospital [35], or in Ontario, with the Quality Standard on Transitions between Hospital and Home [36]. Models of care and systematic healthcare processes must enable interprofessional trust, collaboration, and communication [37]. This is especially the case in stressful, crisis responses, such as during the COVID-19 pandemic and its aftermath [38]. A qualitative study of US healthcare providers’ (n = 18) experiences of interprofessional collaboration during the COVID-19 pandemic (more the acute vs. long COVID experience) revealed that collaboration was promoted with co-location when timely, accurate, and transparent communication was prioritized [39,40]. Interprofessional teams were fragmented by the pandemic and called for leadership support and organizational responses [39,40]. Caring for persons with long COVID necessarily involves the pandemic aftermath, including staff burnout as well as the navigation of ever-changing information and understanding of novel conditions and their sequelae. 

The noted concerns on the reliability and validity of the PCFS were not unfounded, they but must be considered in light of the crisis environment during which the PCFS was identified as needed and developed as a screening (not assessment) tool. A rapid scoping review (n = 10 studies, from 389 screened articles) completed in July 2021 (in the period of this study’s interviews) revealed limited evidence on the psychometric validation of long COVID screening tools generally [41]. Three studies mentioned the PCFS: (1) to test its psychometric validity [42]; (2) to use (but did not psychometrically test) [43]; and (3) to describe it as neither validated nor useable [41,43]. In the study that sought to assess the construct validity of the PCFS (n = 1939), the PCFS was compared to validated tools measuring quality of life (EQ-5D-5L) and work productivity (WPA questionnaire) [34]. There were weak-to-strong statistical associations between PCFS functional status and all domains of the EQ-5D-5L (r: 0.233–0.661) [42]. The WPA findings suggested a reciprocal relationship between WPA increases in activity impairment and PCFS decreases in functional status [34]. This construct validity study was limited as it lacked power to detect “small but meaningful” differences [42]. Construct validity does not inform the reliability of a tool. 

A 2022 study (n = 121) sought to clarify the correlation between the PCFS and quality of life (SF-36), mental health (Hospital Anxiety and Depression Scale), dyspnea (modified Medical Research Council Scale (mMRC)), and functional measure (6 min walk test and Borg dyspnea rating) [44]. There were significant correlations between the PCFS and the tools measuring quality of life (rho = −0.71; *p* < 0.0001); dyspnea (mMRC rho = 0.53, *p* < 0.0001); mental health (rho = 0.39, *p* < 0.0001); function via 6 min walk test (rho = 0.48, *p* < 0.0001); and function via Borg dyspnea rating (rho = 0.48, *p* < 0.0001) [44]. The evidence on the construct validity of the PCFS is growing, but a full psychometric evaluation, including clarification of the reliability and responsiveness, is lacking, but it is necessary to ensure tool robustness [35,45]. This lack of validity, reliability, and responsiveness data would apply to most long-COVID-specific psychometric tools.

This study has limitations. First, the provider experience interviews targeted front-line providers (mostly nursing and allied health professionals) who were implementing some or all of the pathway, which may have limited the perspectives of other professional disciplines or providers unfamiliar with the pathway. We also had limited numbers in each group so could not explore the differences and similarities between different healthcare professionals. However, early-user sites represented a diversity of sites both geographically and professionally, which allowed for a better understanding of common barriers and facilitators amongst care providers with greater knowledge of long COVID care and the pathway. Second, not all early-user sites had individuals participating in both the structured and semi-structured interviews, which means some of the structured findings may not be transferable to all sites. However, this study sought maximum variation in geographical and care continuum representation and to identify a comprehensive understanding of the barriers and facilitators to pathway implementation, not a generalizable experience across sites.

## 5. Conclusions

The perceptions of providers at early-user sites demonstrated facilitators and barriers to implementing a novel provincial long COVID pathway across a diverse provincial healthcare landscape. Perceived benefits of the rehabilitation pathway include providing process benchmarks, offering a breadth of information on an emerging condition, and raising awareness of myriad symptoms and strategies. However, barriers perceived by the providers led to variable adoption of the rehabilitation pathway, as providers were overwhelmed and unable to manage the information onslaught, as well as misunderstandings of the aims of using the pathway’s components. The confusion between using the PCFS as an assessment tool, instead of a screening tool, was particularly concerning. 

Implementation of a rehabilitation pathway developed during a crisis to help support persons with a novel chronic condition in the post-crisis setting is fraught with barriers. However, there are opportunities to advance equity, co-design, and clinical support through researching provider perspectives and then addressing the recognized barriers. Of particular note, involving front-line clinicians in designing or redesigning the pathway is critical.

## 6. Future Directions

Future research directions should center around better understanding how to nimbly implement rehabilitation pathways in crisis settings, including which facilitators can be spread and what can overcome the noted barriers to advance the adoption and feasibility of the pathway. Quantitative data to understand how this implementation experience translated into patient access and outcomes would offer insights into the impact of the long COVID rehabilitation pathway. Follow-up research using hybrid effectiveness-implementation trials may provide robust data that would enable tailored implementation by demonstrating the critical versus adaptable facets of the pathway.

## Figures and Tables

**Table 1 healthcare-12-00730-t001:** Early-user site implementation, monitoring, and reporting of the Post-COVID-19 Functional Scale (PCFS) (i.e., screening tool in provincial pathway).

	Provincial, Outpatient Tele-Rehabilitation (n = 1)	Primary Care (n = 4)	Outpatient Rehabilitation—Metro Area 1 (n = 1)	Outpatient Rehabilitation—Metro Area 2 (n = 1)	Medical Specialty Long COVID Clinic (n = 2)
**Currently Using PCFS**	Yes	Partially	Use PCFS as a triage tool, no PCFS assessment on site	No	Yes
**Intend to Use PCFS**	N/A	N/A	N/A	Yes, possibly for tele-rehabilitation and metro-area 2 medical specialty long COVID clinic referrals	N/A
**Personnel Administering PCFS**	Either by administrative staff on COVID-19 call backs or clinician (OT, PT)	No systematic use of PCFS at clinics, up to individual discretion of providers	N/A	Most likely professional staff that administer other rehab assessments	Either nurse or nurse practitioner
**Storage of PCFS data**	Built into EMR from Sept 2021 onwards	Anticipate storing in EMR (specialist link pathway seems identical to PCFS)	N/A	Potentially in EMR Or in a separate Excel tracking sheet for post-COVID-19 referrals	Site 1 uses both paper and electronic charts. Site 2 uses paper charts, will transition to EMR after May 2022

OT: occupational therapy; PT: physical therapy; EMR: electronic medical record.

## Data Availability

The data presented in this study are available on request from the corresponding author due to privacy considerations.

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
