# Peer review of "Implementation of a Provincial Long COVID Care Pathway in Alberta, Canada: Provider Perceptions"

_healthcare, 2024, doi:10.3390/healthcare12070730_

Round 1
Reviewer 1 Report
Comments and Suggestions for Authors
The authors discuss the important issue of treating the long-term effects of SARS CoV-2 virus infection that health care workers have to face. The article is well organized, but I would like to clarify a few issues.
In the methodological part, please indicate which of the researchers conducted semi-structured and structured interviews? Were they the same researchers who analyzed the collected material? What were the researchers' professions? Did they have previous experience collecting this type of data?
The authors state that they used thematic analysis of qualitative data (they do not provide references on the methodology used).
How many people were offered to participate in the study? How many people refused to participate? For what reasons?
What important sociodemographic characteristics did the respondents present? e.g. age, gender? Did the authors collect such data? if not, why?
Once again, I would like to thank the authors for taking up this topic, I hope that I will be able to re-evaluate the manuscript.
Author Response
TITLE: Implementation of a Provincial Long COVID Care Pathways in Alberta, Canada: Provider Perceptions
RESPONSE TO REVIEWER 1.
Reviewer Comment #1: In the methodological part, please indicate which of the researchers conducted semi-structured and structured interviews? Were they the same researchers who analyzed the collected material? What were the researchers' professions? Did they have previous experience collecting this type of data?
Response to Reviewer Comment:
|
Original Version (location) |
Revised Version |
|
First, structured interviews with site leads demarcated process and implementation characteristics of the site context. Second, semi-structured interviews with front-line providers (1:1 and group) clarified provider perspectives on interim implementation outcomes of the long COVID rehabilitation pathway. (ln 149-155) |
First, structured interviews (by researchers SH or JK) with site leads demarcated process and implementation characteristics of the site context. Second, semi-structured interviews (by researchers SH or JK) with front-line providers (1:1 and group) clarified provider perspectives on interim implementation outcomes of the long COVID rehabilitation pathway. Both researchers SH and JK are health-research trainees, with previous education and experience in trained qualitative research data collection and analysis. |
Reviewer Comment #2: The authors state that they used thematic analysis of qualitative data (they do not provide references on the methodology used).
Response to Reviewer Comment:
|
Original Version (location) |
Revised Version |
|
Qualitative content analysis was employed to identify commonalities and differences in the data. (ln185-186)
Discussion (ln 480-530) The pathway developed likely requires greater emphasis and strategies on interprofessional communication and collaboration to empower teams to share information and to trust that shared information. This could reduce duplication of screening conducted on patients with significant fatigue issues; allay concerns about patients falling through the cracks; and foster the standardization of care and access across diverse sites. A consistent health system challenge is the quality of transitions across the care continuum. This is reflected in organizational, health system priorities such as in Alberta, that prioritizes patient flow [39] and transitions between home and hospital [40]; or in Ontario with the Quality Standard on Transitions between Hospital and Home [41]. Models of care and systematic healthcare processes must enable interprofessional trust, collaboration and communication [42]. This is especially the case in stressful, crisis responses, such as during the COVID-19 pandemic and its aftermath [43]. A qualitative study of US healthcare providers’ (n=18) experience of interprofessional collaboration during COVID-19 pandemic (more the acute vs. long COVID experience) revealed that collaboration was promoted with co-location, when timely, accurate, and transparent communication was prioritized [43]. Interprofessional teams were fragmented by the pandemic and called for leadership support and organizational responses [43]. Caring for persons with long COVID necessarily involves the pandemic aftermath, including staff burnout as well as the navigation of ever-changing information and understanding of novel conditions and their sequelae. The noted concerns on the reliability and validity of the PCFS were not unfounded, but must be considered in light of the crisis environment during which the PCFS was identified as needed and developed as a screening (not assessment) tool. A rapid scoping review (n=10 studies, from 389 screened articles) completed in July 2021 (in the period of the study interviews) revealed limited evidence on the psychometric validation of long COVID screening tools generally [33]. Three studies mentioned the PCFS: (1) to test its psychometric validity [34]; (2) to use (but did not psychometrically test) [35]; and (3) to describe it as neither validated nor useable [33, 36]. In the study that sought to assess the construct validity of the PCFS (n=1939), the PCFS was compared to validated tools measuring quality of life (EQ-5D-5L) and work productivity (WPA questionnaire) [34]. There were weak-to-strong statistical associations between PCFS functional status and all domains of the EQ-5D-5L (r:0.233-0.661) [34]. The WPA findings suggested a reciprocal relationship between WPA increases of activity impairment and PCFS decreases in functional status [34]. This construct-validity study was limited as it lacked power to detect “small but meaningful” differences [34]. Construct validity does not inform the reliability of a tool. A 2022 study (n=121) sought to clarify the correlation between the PCFS and quality of life (SF-36), mental health (Hospital Anxiety and Depression Scale), dyspnea (modified Medical Research Council Scale (mMRC)), and functional measure (6-minute walk test and Borg Dyspnea rating) [37]. There were significant correlations between the PCFS and the tools measuring quality of life (rho=-0.71; p<0.0001); dyspnea (mMRC rho=0.53, p<0.0001); mental health (rho=0.39, p<0.0001); function via 6-minute walk test (rho=0.48, p<0.0001) and function via Borg dyspnea rating (rho=0.48, p<0.0001) [37]. The evidence on the construct validity of the PCFS is growing, but a full psychometric evaluation including clarification of the reliability and responsiveness is lacking and necessary to ensure tool robustness [38]. This lack of validity, reliability and responsiveness data would apply to most long-COVID-specific psychometric tools.
32. Michie, S., et al., Making psychological theory useful for implementing evidence based practice: A consensus approach. Quality and Safety in Health Care, 2005. 14(1): p. 26-33. 33. Scientific Advisory Group, Rapid Review: Chronic Symptoms of COVID-19. 2021, Alberta Health Services: Edmonton, AB. p. 1-120. 34. Machado, F.V.C., et al., Construct validity of the Post-COVID-19 Functional Status Scale in adult subjects with COVID-19. Health Qual Life Outcomes, 2021. 19(1): p. 40. 35. D'Cruz, R.F., et al., Chest radiography is a poor predictor of respiratory symptoms and functional impairment in survivors of severe COVID-19 pneumonia. ERJ Open Research, 2021. 7(1): p. 655-2020. 36. Vehar, S., et al., Post-acute sequelae of SARS-CoV-2 infection: Caring for the 'long-haulers'. Cleveland Clinic journal of medicine, 2021. 37. Benkalfate, N., et al., Evaluation of the Post-COVID-19 Functional Status (PCFS) Scale in a cohort of patients recovering from hypoxemic SARS-CoV-2 pneumonia. BMJ Open Respir Res, 2022. 9(1). 38. De Vet, H.C.W., et al., Measurement in Medicine. 2011, Cambridge, UK: Cambridge University Press. 39. Cowell, J., Healthcare Action Plan: 90-Day Report from Dr. John Cowell, Official Administrator. 2022, Government of Alberta: Edmonton, AB. 40. Alberta Health Services;Primary Care Networks;, and Government of Alberta;, Alberta's Home to Hospital to Home: Transition Guidelines. 20202, Alberta Health Services: Edmonton, AB. p. 36. 41. Ontario, H.Q., Transitions Between Hospital and Home Quality Standard. 2018(January). 42. Karam, M., et al., Comparing interprofessional and interorganizational collaboration in healthcare: A systematic review of the qualitative research. Int J Nurs Stud, 2018. 79: p. 70-83. 43. Jordan, S.R., S.C. Connors, and K.A. Mastalerz, Frontline healthcare workers' perspectives on interprofessional teamwork during COVID-19. J Interprof Educ Pract, 2022. 29: p. 100550. (ln 646-667) |
Qualitative content analysis was employed to identify commonalities and differences in the data [33, 34, 35]. (ln 186)
Discussion (ln 480-533) The pathway developed likely requires greater emphasis and strategies on interprofessional communication and collaboration to empower teams to share information and to trust that shared information. This could reduce duplication of screening conducted on patients with significant fatigue issues; allay concerns about patients falling through the cracks; and foster the standardization of care and access across diverse sites. A consistent health system challenge is the quality of transitions across the care continuum. This is reflected in organizational, health system priorities such as in Alberta, that prioritizes patient flow [40] and transitions between home and hospital [41]; or in Ontario with the Quality Standard on Transitions between Hospital and Home [42]. Models of care and systematic healthcare processes must enable interprofessional trust, collaboration and communication [43]. This is especially the case in stressful, crisis responses, such as during the COVID-19 pandemic and its aftermath [44]. A qualitative study of US healthcare providers’ (n=18) experience of interprofessional collaboration during COVID-19 pandemic (more the acute vs. long COVID experience) revealed that collaboration was promoted with co-location, when timely, accurate, and transparent communication was prioritized [44]. Interprofessional teams were fragmented by the pandemic and called for leadership support and organizational responses [44]. Caring for persons with long COVID necessarily involves the pandemic aftermath, including staff burnout as well as the navigation of ever-changing information and understanding of novel conditions and their sequelae. The noted concerns on the reliability and validity of the PCFS were not unfounded, but must be considered in light of the crisis environment during which the PCFS was identified as needed and developed as a screening (not assessment) tool. A rapid scoping review (n=10 studies, from 389 screened articles) completed in July 2021 (in the period of the study interviews) revealed limited evidence on the psychometric validation of long COVID screening tools generally [34]. Three studies mentioned the PCFS: (1) to test its psychometric validity [35]; (2) to use (but did not psychometrically test) [36]; and (3) to describe it as neither validated nor useable [34, 37]. In the study that sought to assess the construct validity of the PCFS (n=1939), the PCFS was compared to validated tools measuring quality of life (EQ-5D-5L) and work productivity (WPA questionnaire) [35]. There were weak-to-strong statistical associations between PCFS functional status and all domains of the EQ-5D-5L (r:0.233-0.661) [35]. The WPA findings suggested a reciprocal relationship between WPA increases of activity impairment and PCFS decreases in functional status [35]. This construct-validity study was limited as it lacked power to detect “small but meaningful” differences [35]. Construct validity does not inform the reliability of a tool. A 2022 study (n=121) sought to clarify the correlation between the PCFS and quality of life (SF-36), mental health (Hospital Anxiety and Depression Scale), dyspnea (modified Medical Research Council Scale (mMRC)), and functional measure (6-minute walk test and Borg Dyspnea rating) [38]. There were significant correlations between the PCFS and the tools measuring quality of life (rho=-0.71; p<0.0001); dyspnea (mMRC rho=0.53, p<0.0001); mental health (rho=0.39, p<0.0001); function via 6-minute walk test (rho=0.48, p<0.0001) and function via Borg dyspnea rating (rho=0.48, p<0.0001) [38]. The evidence on the construct validity of the PCFS is growing, but a full psychometric evaluation including clarification of the reliability and responsiveness is lacking and necessary to ensure tool robustness [39]. This lack of validity, reliability and responsiveness data would apply to most long-COVID-specific psychometric tools.
References: 33. Sandelowski, M. What’s in a name? Qualitative description revisited. Res. Nurs. Health 2009, 33, 77–84. 34. Hsieh, H.-F.; Shannon, S.E. Three Approaches to Qualitative Content Analysis. Qual. Health Res. 2005, 15, 1277–1288. 35. Milne, J.; Oberle, K. Enhancing Rigor in Qualitative Description: A case study. J. Wound Ostomy Cont. Nurs. 2005, 32, 413–42. 36. Scientific Advisory Group, Rapid Review: Chronic Symptoms of COVID-19. 2021, Alberta Health Services: Edmonton, AB. p. 1-120. 37. Machado, F.V.C., et al., Construct validity of the Post-COVID-19 Functional Status Scale in adult subjects with COVID-19. Health Qual Life Outcomes, 2021. 19(1): p. 40. 38. D'Cruz, R.F., et al., Chest radiography is a poor predictor of respiratory symptoms and functional impairment in survivors of severe COVID-19 pneumonia. ERJ Open Research, 2021. 7(1): p. 655-2020. 39. Vehar, S., et al., Post-acute sequelae of SARS-CoV-2 infection: Caring for the 'long-haulers'. Cleveland Clinic journal of medicine, 2021. 40. Benkalfate, N., et al., Evaluation of the Post-COVID-19 Functional Status (PCFS) Scale in a cohort of patients recovering from hypoxemic SARS-CoV-2 pneumonia. BMJ Open Respir Res, 2022. 9(1). 41. De Vet, H.C.W., et al., Measurement in Medicine. 2011, Cambridge, UK: Cambridge University Press. 42. Cowell, J., Healthcare Action Plan: 90-Day Report from Dr. John Cowell, Official Administrator. 2022, Government of Alberta: Edmonton, AB. 43. Alberta Health Services;Primary Care Networks;, and Government of Alberta;, Alberta's Home to Hospital to Home: Transition Guidelines. 20202, Alberta Health Services: Edmonton, AB. p. 36. 44. Ontario, H.Q., Transitions Between Hospital and Home Quality Standard. 2018(January). 45. Karam, M., et al., Comparing interprofessional and interorganizational collaboration in healthcare: A systematic review of the qualitative research. Int J Nurs Stud, 2018. 79: p. 70-83. 46. Jordan, S.R., S.C. Connors, and K.A. Mastalerz, Frontline healthcare workers' perspectives on interprofessional teamwork during COVID-19. J Interprof Educ Pract, 2022. 29: p. 100550. (ln 650-673) |
Reviewer Comment #3: How many people were offered to participate in the study? How many people refused to participate? For what reasons?
Response to Reviewer Comment:
|
Original Version (location) |
Revised Version |
|
3.2. Participants Between January and August 2022, eight early-user sites supported study recruitment from their teams leading to five structured, team-lead interviews (n=13 participants; four group and one 1:1 interview) and seven semi-structured, multidisciplinary, front-line clinician interviews (n=15 participants; five group and two 1:1 interviews). These sites represented primary care (n=4), outpatient rehabilitation (n=3), and COVID-19 specialty clinics (n=2) (Table 1). The structured and semi-structured interviews ranged from 30-60 and 20-60 minutes, respectively. Not all early-user sites participated in both the structured and unstructured interviews. In the structured interviews, the role participants primarily identified with included team leads (n=6), senior consultants/advisors (n=2), physiotherapists (n=2), director (n=1), physician (n=1), and an occupational therapist (n=1). The professions represented in the semi-structured interviews included nursing, pharmacy, physiotherapy, occupational therapy, and recreation therapy. (ln 213-226) |
3.2. Participants Between January and August 2022, eight early-user sites supported study recruitment from their teams leading to five structured, team-lead interviews (n=13 participants; four group and one 1:1 interview) and seven semi-structured, multidisciplinary, front-line clinician interviews (n=15 participants; five group and two 1:1 interviews). These sites represented primary care (n=4), outpatient rehabilitation (n=3), and COVID-19 specialty clinics (n=2) (Table 1). The structured and semi-structured interviews ranged from 30-60 and 20-60 minutes, respectively. Not all early-user sites participated in both the structured and unstructured interviews. In the structured interviews, the role participants primarily identified with included team leads (n=6), senior consultants/advisors (n=2), physiotherapists (n=2), director (n=1), physician (n=1), and an occupational therapist (n=1). The professions represented in the semi-structured interviews included nursing, pharmacy, physiotherapy, occupational therapy, and recreation therapy. Because recruitment was facilitated through intermediaries (team leads) to protect clinician privacy, we are unable to determine how many people declined study participation and their reasons therein. (ln 213-226) |
Reviewer Comment #4: What important sociodemographic characteristics did the respondents present? e.g. age, gender? Did the authors collect such data? if not, why?
Response to Reviewer Comment:
|
Original Version (location) |
Revised Version |
|
2.2.2. Semi-Structured Interviews The data retrieved from closed-ended questions complemented, and provided context for, the open-ended question responses. Informed by the structured interviews, virtual individual or group semi-structured interviews were completed with consenting front-line clinicians (Supplemental Table 2). Open-ended questions queried current roles and responsibilities; current approaches and challenges to screening and assessing for long COVID symptoms; training and perceptions of the PCFS (as a proxy of framework use); familiarity and understanding of the provincial long COVID rehabilitation pathway; familiarity and understanding of available educational resources for patients or providers; and perceived acceptability, adoption, feasibility and fidelity of the pathway. All semi-structured interviews were audio-recorded and confidentially transcribed verbatim. Written field notes were recorded during the interview. Ln (169-180) |
2.2.2. Semi-Structured Interviews The data retrieved from closed-ended questions complemented, and provided context for, the open-ended question responses. Informed by the structured interviews, virtual individual or group semi-structured interviews were completed with consenting front-line clinicians (Supplemental Table 2). Open-ended questions queried current roles and responsibilities; current approaches and challenges to screening and assessing for long COVID symptoms; training and perceptions of the PCFS (as a proxy of framework use); familiarity and understanding of the provincial long COVID rehabilitation pathway; familiarity and understanding of available educational resources for patients or providers; and perceived acceptability, adoption, feasibility and fidelity of the pathway. All semi-structured interviews were audio-recorded and confidentially transcribed verbatim. Written field notes were recorded during the interview. Due to limited time with clinician-participants, demographic questions had to be removed from the structured and semi-structured interview guides. (ln 169-182)
|

Reviewer 2 Report
Comments and Suggestions for Authors
Congrats for the worked, long COVID-19, it´s a new impact disease.
All the recommendations are to improve the content of research.
Abstract- It´s not clear the methodology used to do the content analysis, for example Bardin´s methods.
Materials & Methods-It is not clear why this number of participants has reached data saturation?
Why you used two differents kind of interview and analised then as equal?
What method of analysis is used, given that the programs are not scientific methods.
"the care continuum[JK1]" Is it wrong reference?
Author Response
TITLE: Implementation of a Provincial Long COVID Care Pathways in Alberta, Canada: Provider Perceptions
RESPONSE TO REVIEWER 2.
Reviewer Comment #1: Abstract- It´s not clear the methodology used to do the content analysis, for example Bardin´s methods.
Response to Reviewer Comment:
|
Original Version (location) |
Revised Version |
|
Abstract: A novel, complex chronic condition emerged from the COVID-19 pandemic: long COVID. The persistent long COVID symptoms can be multisystem and varied. Effective long COVID management requires multidisciplinary, collaborative models of care, which continue to be developed and refined. Alberta’s provincial health system developed a novel long COVID pathway. We aimed to clarify the perspectives of multidisciplinary healthcare providers on the early implementation of the provincial long COVID pathway, particularly pathway acceptability, adoption, feasibility, and fidelity. Provider-participants were recruited from eight early-user sites from across the care continuum. Sites represented primary care (n=4), outpatient rehabilitation (n=3), and COVID-19 specialty clinics (n=2). Participants participated in structured or semi-structured virtual interviews (both group or 1:1 were available). Structured interviews sought to clarify context, processes, and pathway use; semi-structured interviews targeted provider perceptions of pathway implementation including barriers and facilitators. Across the eight sites that participated, five structured interviews (n=13 participants) and seven semi-structured interviews (n=15 participants) were completed. Sites represented primary care (n=4), outpatient rehabilitation (n=3), and COVID-19 specialty clinics (n=2). Qualitative content analysis was used on transcripts and field notes. Provider perceptions of the early implementation outcomes of the provincial long COVID pathway revealed three key themes: process perceptions; awareness of patient educational resources; and challenges of evolving knowledge. (ln23-40)
|
Abstract: A novel, complex chronic condition emerged from the COVID-19 pandemic: long COVID. The persistent long COVID symptoms can be multisystem and varied. Effective long COVID management requires multidisciplinary, collaborative models of care, which continue to be developed and refined. Alberta’s provincial health system developed a novel long COVID pathway. We aimed to clarify the perspectives of multidisciplinary healthcare providers on the early implementation of the provincial long COVID pathway, particularly pathway acceptability, adoption, feasibility, and fidelity using Sandelowki’s qualitative description. Provider-participants were recruited from eight early-user sites from across the care continuum. Sites represented primary care (n=4), outpatient rehabilitation (n=3), and COVID-19 specialty clinics (n=2). Participants participated in structured or semi-structured virtual interviews (both group or 1:1 were available). Structured interviews sought to clarify context, processes, and pathway use; semi-structured interviews targeted provider perceptions of pathway implementation including barriers and facilitators. Analysis was guided by Hseih & Shannon as well as Sandelowski. Across the eight sites that participated, five structured interviews (n=13 participants) and seven semi-structured interviews (n=15 participants) were completed. Sites represented primary care (n=4), outpatient rehabilitation (n=3), and COVID-19 specialty clinics (n=2). Qualitative content analysis was used on transcripts and field notes. Provider perceptions of the early implementation outcomes of the provincial long COVID pathway revealed three key themes: process perceptions; awareness of patient educational resources; and challenges of evolving knowledge. (ln 23-41)
|
Reviewer Comment #2: Materials & Methods-It is not clear why this number of participants has reached data saturation?
Response to Reviewer Comment:
|
Original Version (location) |
Revised Version |
|
Study participants included multidisciplinary healthcare providers working at early-user study sites. Role-wise, we sought maximum variation across frontline clinical staff, clinical team leads, site managers and leadership. Discipline-wise, we sought maximum variation across medicine, nursing, occupational therapy, physiotherapy, and other allied health disciplines. There were no exclusion criteria for providers. (ln137-141) |
Study participants included multidisciplinary healthcare providers working at early-user study sites. Role-wise, we sought maximum variation across frontline clinical staff, clinical team leads, site managers and leadership. Discipline-wise, we sought maximum variation across medicine, nursing, occupational therapy, physiotherapy, and other allied health disciplines. There were no exclusion criteria for providers. Data collection was planned until data saturation, or when three repeated attempts at recruitment at the same site yielded no further participants.
(ln 137-144) |
Reviewer Comment #3: Why you used two differents kind of interview and analised then as equal?
Response to Reviewer Comment:
|
Original Version (location) |
Revised Version |
|
2.3. Data Analysis Structured responses, transcripts, and field notes for both the structured and semi-structured interviews were collected and analyzed concurrently using Microsoft Excel and NVivo 12, respectively. An audit trail was kept of decision-making throughout the interview and analysis process. The structured responses were descriptively analyzed with a focus on frequencies, distributions, and key contextual practices. Qualitative content analysis was employed to identify commonalities and differences in the data. Field notes were inductively coded to identify sub-categories within these broad categories. Semi-structured interview transcripts were analyzed thematically. One interviewer (SH) and one independent member of the research team (KB) independently coded each interview. A third researcher (JAK) independently assessed the coding to resolve discrepancies and ensure alignment between codes and transcripts. The researchers did not have previous connections with the participants. Team members met regularly to develop and review the coding framework, which sought to expose relationships between themes and to produce a holistic understanding of the participants’ perspectives. Thick description with detailed participant quotes substantiated the coding framework. (ln 177-203) |
2.3. Data Analysis Structured responses, transcripts, and field notes for both the structured and semi-structured interviews were collected and analyzed concurrently using Microsoft Excel and NVivo 12, respectively. The different data sources were deemed complementary, offering contextual insights or content details respectively. An audit trail was kept of decision-making throughout the interview and analysis process. The structured responses were descriptively analyzed with a focus on frequencies, distributions, and key contextual practices. Meanwhile, qualitative content analysis was employed to identify commonalities and differences in the semi-structured transcripts and field notes [33, 34, 35]. Field notes were inductively coded to identify sub-categories within these broad categories. Semi-structured interview transcripts were analyzed thematically. One interviewer (SH) and one independent member of the research team (KB) independently coded each interview. A third researcher (JAK) independently assessed the coding to resolve discrepancies and ensure alignment between codes and transcripts. The researchers did not have previous connections with the participants. Team members met regularly to develop and review the coding framework, which sought to expose relationships between themes and to produce a holistic understanding of the participants’ perspectives. Thick description with detailed participant quotes substantiated the coding framework. (ln 196-203
|
Reviewer Comment #4: What method of analysis is used, given that the programs are not scientific methods.
Response to Reviewer Comment:
Thanks for this comment. Unfortunately, it is unclear what is meant by the comment. The first half of the question (“What method of analysis is used”) suggests a query around the qualitative methodology. This was noted by reviewer #1, and we have added the changes noted in the table below. Please note that the length of the changes is due to updating all the subsequent references. However, the second half of the question “, given that the programs are not scientific methods” adds confusion as it seems to speak to the clinical long COVID programs? We would be happy to address this comment if we could be advised as to which section/area the reviewer is referring to. Or, let us know if the below suffices.
Response to Reviewer Comment:
|
Original Version (location) |
Revised Version |
|
Qualitative content analysis was employed to identify commonalities and differences in the data. (ln185-186)
Discussion (ln 480-530) The pathway developed likely requires greater emphasis and strategies on interprofessional communication and collaboration to empower teams to share information and to trust that shared information. This could reduce duplication of screening conducted on patients with significant fatigue issues; allay concerns about patients falling through the cracks; and foster the standardization of care and access across diverse sites. A consistent health system challenge is the quality of transitions across the care continuum. This is reflected in organizational, health system priorities such as in Alberta, that prioritizes patient flow [39] and transitions between home and hospital [40]; or in Ontario with the Quality Standard on Transitions between Hospital and Home [41]. Models of care and systematic healthcare processes must enable interprofessional trust, collaboration and communication [42]. This is especially the case in stressful, crisis responses, such as during the COVID-19 pandemic and its aftermath [43]. A qualitative study of US healthcare providers’ (n=18) experience of interprofessional collaboration during COVID-19 pandemic (more the acute vs. long COVID experience) revealed that collaboration was promoted with co-location, when timely, accurate, and transparent communication was prioritized [43]. Interprofessional teams were fragmented by the pandemic and called for leadership support and organizational responses [43]. Caring for persons with long COVID necessarily involves the pandemic aftermath, including staff burnout as well as the navigation of ever-changing information and understanding of novel conditions and their sequelae. The noted concerns on the reliability and validity of the PCFS were not unfounded, but must be considered in light of the crisis environment during which the PCFS was identified as needed and developed as a screening (not assessment) tool. A rapid scoping review (n=10 studies, from 389 screened articles) completed in July 2021 (in the period of the study interviews) revealed limited evidence on the psychometric validation of long COVID screening tools generally [33]. Three studies mentioned the PCFS: (1) to test its psychometric validity [34]; (2) to use (but did not psychometrically test) [35]; and (3) to describe it as neither validated nor useable [33, 36]. In the study that sought to assess the construct validity of the PCFS (n=1939), the PCFS was compared to validated tools measuring quality of life (EQ-5D-5L) and work productivity (WPA questionnaire) [34]. There were weak-to-strong statistical associations between PCFS functional status and all domains of the EQ-5D-5L (r:0.233-0.661) [34]. The WPA findings suggested a reciprocal relationship between WPA increases of activity impairment and PCFS decreases in functional status [34]. This construct-validity study was limited as it lacked power to detect “small but meaningful” differences [34]. Construct validity does not inform the reliability of a tool. A 2022 study (n=121) sought to clarify the correlation between the PCFS and quality of life (SF-36), mental health (Hospital Anxiety and Depression Scale), dyspnea (modified Medical Research Council Scale (mMRC)), and functional measure (6-minute walk test and Borg Dyspnea rating) [37]. There were significant correlations between the PCFS and the tools measuring quality of life (rho=-0.71; p<0.0001); dyspnea (mMRC rho=0.53, p<0.0001); mental health (rho=0.39, p<0.0001); function via 6-minute walk test (rho=0.48, p<0.0001) and function via Borg dyspnea rating (rho=0.48, p<0.0001) [37]. The evidence on the construct validity of the PCFS is growing, but a full psychometric evaluation including clarification of the reliability and responsiveness is lacking and necessary to ensure tool robustness [38]. This lack of validity, reliability and responsiveness data would apply to most long-COVID-specific psychometric tools.
32. Michie, S., et al., Making psychological theory useful for implementing evidence based practice: A consensus approach. Quality and Safety in Health Care, 2005. 14(1): p. 26-33. 33. Scientific Advisory Group, Rapid Review: Chronic Symptoms of COVID-19. 2021, Alberta Health Services: Edmonton, AB. p. 1-120. 34. Machado, F.V.C., et al., Construct validity of the Post-COVID-19 Functional Status Scale in adult subjects with COVID-19. Health Qual Life Outcomes, 2021. 19(1): p. 40. 35. D'Cruz, R.F., et al., Chest radiography is a poor predictor of respiratory symptoms and functional impairment in survivors of severe COVID-19 pneumonia. ERJ Open Research, 2021. 7(1): p. 655-2020. 36. Vehar, S., et al., Post-acute sequelae of SARS-CoV-2 infection: Caring for the 'long-haulers'. Cleveland Clinic journal of medicine, 2021. 37. Benkalfate, N., et al., Evaluation of the Post-COVID-19 Functional Status (PCFS) Scale in a cohort of patients recovering from hypoxemic SARS-CoV-2 pneumonia. BMJ Open Respir Res, 2022. 9(1). 38. De Vet, H.C.W., et al., Measurement in Medicine. 2011, Cambridge, UK: Cambridge University Press. 39. Cowell, J., Healthcare Action Plan: 90-Day Report from Dr. John Cowell, Official Administrator. 2022, Government of Alberta: Edmonton, AB. 40. Alberta Health Services;Primary Care Networks;, and Government of Alberta;, Alberta's Home to Hospital to Home: Transition Guidelines. 20202, Alberta Health Services: Edmonton, AB. p. 36. 41. Ontario, H.Q., Transitions Between Hospital and Home Quality Standard. 2018(January). 42. Karam, M., et al., Comparing interprofessional and interorganizational collaboration in healthcare: A systematic review of the qualitative research. Int J Nurs Stud, 2018. 79: p. 70-83. 43. Jordan, S.R., S.C. Connors, and K.A. Mastalerz, Frontline healthcare workers' perspectives on interprofessional teamwork during COVID-19. J Interprof Educ Pract, 2022. 29: p. 100550. (ln 646-667) |
Qualitative content analysis was employed to identify commonalities and differences in the data [33, 34, 35]. (ln 186)
Discussion (ln 480-533) The pathway developed likely requires greater emphasis and strategies on interprofessional communication and collaboration to empower teams to share information and to trust that shared information. This could reduce duplication of screening conducted on patients with significant fatigue issues; allay concerns about patients falling through the cracks; and foster the standardization of care and access across diverse sites. A consistent health system challenge is the quality of transitions across the care continuum. This is reflected in organizational, health system priorities such as in Alberta, that prioritizes patient flow [40] and transitions between home and hospital [41]; or in Ontario with the Quality Standard on Transitions between Hospital and Home [42]. Models of care and systematic healthcare processes must enable interprofessional trust, collaboration and communication [43]. This is especially the case in stressful, crisis responses, such as during the COVID-19 pandemic and its aftermath [44]. A qualitative study of US healthcare providers’ (n=18) experience of interprofessional collaboration during COVID-19 pandemic (more the acute vs. long COVID experience) revealed that collaboration was promoted with co-location, when timely, accurate, and transparent communication was prioritized [44]. Interprofessional teams were fragmented by the pandemic and called for leadership support and organizational responses [44]. Caring for persons with long COVID necessarily involves the pandemic aftermath, including staff burnout as well as the navigation of ever-changing information and understanding of novel conditions and their sequelae. The noted concerns on the reliability and validity of the PCFS were not unfounded, but must be considered in light of the crisis environment during which the PCFS was identified as needed and developed as a screening (not assessment) tool. A rapid scoping review (n=10 studies, from 389 screened articles) completed in July 2021 (in the period of the study interviews) revealed limited evidence on the psychometric validation of long COVID screening tools generally [34]. Three studies mentioned the PCFS: (1) to test its psychometric validity [35]; (2) to use (but did not psychometrically test) [36]; and (3) to describe it as neither validated nor useable [34, 37]. In the study that sought to assess the construct validity of the PCFS (n=1939), the PCFS was compared to validated tools measuring quality of life (EQ-5D-5L) and work productivity (WPA questionnaire) [35]. There were weak-to-strong statistical associations between PCFS functional status and all domains of the EQ-5D-5L (r:0.233-0.661) [35]. The WPA findings suggested a reciprocal relationship between WPA increases of activity impairment and PCFS decreases in functional status [35]. This construct-validity study was limited as it lacked power to detect “small but meaningful” differences [35]. Construct validity does not inform the reliability of a tool. A 2022 study (n=121) sought to clarify the correlation between the PCFS and quality of life (SF-36), mental health (Hospital Anxiety and Depression Scale), dyspnea (modified Medical Research Council Scale (mMRC)), and functional measure (6-minute walk test and Borg Dyspnea rating) [38]. There were significant correlations between the PCFS and the tools measuring quality of life (rho=-0.71; p<0.0001); dyspnea (mMRC rho=0.53, p<0.0001); mental health (rho=0.39, p<0.0001); function via 6-minute walk test (rho=0.48, p<0.0001) and function via Borg dyspnea rating (rho=0.48, p<0.0001) [38]. The evidence on the construct validity of the PCFS is growing, but a full psychometric evaluation including clarification of the reliability and responsiveness is lacking and necessary to ensure tool robustness [39]. This lack of validity, reliability and responsiveness data would apply to most long-COVID-specific psychometric tools.
References: 33. Sandelowski, M. What’s in a name? Qualitative description revisited. Res. Nurs. Health 2009, 33, 77–84. 34. Hsieh, H.-F.; Shannon, S.E. Three Approaches to Qualitative Content Analysis. Qual. Health Res. 2005, 15, 1277–1288. 35. Milne, J.; Oberle, K. Enhancing Rigor in Qualitative Description: A case study. J. Wound Ostomy Cont. Nurs. 2005, 32, 413–42. 36. Scientific Advisory Group, Rapid Review: Chronic Symptoms of COVID-19. 2021, Alberta Health Services: Edmonton, AB. p. 1-120. 37. Machado, F.V.C., et al., Construct validity of the Post-COVID-19 Functional Status Scale in adult subjects with COVID-19. Health Qual Life Outcomes, 2021. 19(1): p. 40. 38. D'Cruz, R.F., et al., Chest radiography is a poor predictor of respiratory symptoms and functional impairment in survivors of severe COVID-19 pneumonia. ERJ Open Research, 2021. 7(1): p. 655-2020. 39. Vehar, S., et al., Post-acute sequelae of SARS-CoV-2 infection: Caring for the 'long-haulers'. Cleveland Clinic journal of medicine, 2021. 40. Benkalfate, N., et al., Evaluation of the Post-COVID-19 Functional Status (PCFS) Scale in a cohort of patients recovering from hypoxemic SARS-CoV-2 pneumonia. BMJ Open Respir Res, 2022. 9(1). 41. De Vet, H.C.W., et al., Measurement in Medicine. 2011, Cambridge, UK: Cambridge University Press. 42. Cowell, J., Healthcare Action Plan: 90-Day Report from Dr. John Cowell, Official Administrator. 2022, Government of Alberta: Edmonton, AB. 43. Alberta Health Services;Primary Care Networks;, and Government of Alberta;, Alberta's Home to Hospital to Home: Transition Guidelines. 20202, Alberta Health Services: Edmonton, AB. p. 36. 44. Ontario, H.Q., Transitions Between Hospital and Home Quality Standard. 2018(January). 45. Karam, M., et al., Comparing interprofessional and interorganizational collaboration in healthcare: A systematic review of the qualitative research. Int J Nurs Stud, 2018. 79: p. 70-83. 46. Jordan, S.R., S.C. Connors, and K.A. Mastalerz, Frontline healthcare workers' perspectives on interprofessional teamwork during COVID-19. J Interprof Educ Pract, 2022. 29: p. 100550. (ln 650-673) |
Reviewer Comment #5: "the care continuum[JK1]" Is it wrong reference?
Response to Reviewer Comment:
Thanks for this feedback. [JK1] was a comment from a co-author and has been removed from the manuscript.
|
Original Version (location) |
Revised Version |
|
A consistent health system challenge is the quality of transitions across the care continuum. (ln 503-504) |
A consistent health system challenge is the quality of transitions across the care continuum. (ln 503-504) |

Round 2
Reviewer 2 Report
Comments and Suggestions for Authors
Congrats for the revision.